# Mercury Levels in Feathers of Penguins from the Antarctic Peninsula Area: Geographical and Inter-Specific Differences

**DOI:** 10.3390/ijerph18189918

**Published:** 2021-09-21

**Authors:** Miguel Motas, Silvia Jerez, Marta Esteban, Francisco Valera, José Javier Cuervo, Andrés Barbosa

**Affiliations:** 1Área de Toxicología, Facultad de Veterinaria, Universidad de Murcia, Campus de Espinardo, 30100 Murcia, Spain; silviajerez@um.es; 2Área de Toxicología Ambiental, Centro Nacional de Sanidad Ambiental, Instituto de Salud Carlos III, Ctra. Majadahonda-Madrid, 28220 Majadahonda, Spain; m.esteban@isciii.es; 3Departamento de Ecología Funcional y Evolutiva, Estación Experimental de Zonas Áridas, CSIC, Carretera de Sacramento s/n, La Cañada de San Urbano, 04120 Almería, Spain; pvalera@eeza.csic.es (F.V.); jjcuervo@mncn.csic.es (J.J.C.); barbosa@mncn.csic.es (A.B.); 4Departamento de Ecología Evolutiva, Museo Nacional de Ciencias Naturales, CSIC, C/José Gutiérrez Abascal, 2, 28006 Madrid, Spain

**Keywords:** mercury, penguins, feathers, Antarctic Peninsula, biomonitoring

## Abstract

Polar regions, symbols of wilderness, have been identified as potential sinks of mercury coming from natural and anthropogenic sources at lower latitudes. Changes in ice coverage currently occurring in some areas such as the Antarctic Peninsula could enhance these phenomena and their impacts on local biota. As long-lived species at the top of food chains, seabirds are particularly sensitive to this highly toxic metal with the capacity to be biomagnified. Specifically, their feathers can be useful for Hg monitoring since they mainly accumulate its most toxic and persistent form, methyl-Hg. To that end, feathers of gentoo (*Pygoscelis papua*), chinstrap (*P. antarcticus*), and Adélie penguins (*P. adeliae*) (*n* = 108) were collected by passive sampling in seven different locations throughout the Antarctic Peninsula area and analyzed by ICP-MS after microwave-digestion. More than 93% of the samples showed detectable Hg levels (range: 6.3–12,529.8 ng g^−1^ dry weight), and the highest ones were found in the feathers of chinstrap penguins from King George Island. Hg bioconcentration and biomagnification seem to be occurring in the Antarctic food web, giving rise to high but non-toxic Hg levels in penguins, similar to those previously found in Arctic seabirds.

## 1. Introduction

Mercury levels naturally present in the environment have increased during the last decades as a consequence of direct emissions from human activities (industry, combustion of fossil fuels, and solid wastes) and the mobilization of previously deposited mercury [1,2]. This heavy metal has physicochemical properties that confer it special characteristics from the environmental point of view. Mercury is methylated to methylmercury by microorganisms present in aquatic environments, increasing its bioavailability and toxicity [3]. Then, methylmercury is taken up by higher organisms and accumulated and concentrated in the food chain. As a consequence, predators in the high levels of the food chain can be burdened with higher methylmercury levels than those found in water (up to 10^6^–10^7^ times). It has been reported that each change in the food chain accounts for an increase of an order of magnitude in methylmercury concentrations [4]. As a consequence, methylmercury represents less than 1% of mercury in marine and freshwater habitats [5]. Moreover, mercury is not only deposited near its emission sources but can suffer long-range atmospheric transport resulting in significant contamination of terrestrial and aquatic environments and organisms, which constitutes an environmental problem in remote regions in which there are no mercury emissions [6,7]. All this, together with the high toxicity of mercury, which is in the third position in the Agency for Toxic Substances and Disease Registry, US Department of Health and Human Services (ATSDR) priority list of hazardous substances, makes mercury pollution a global contamination problem.

Ambient monitoring allows us to know the levels of environmental chemicals present in different environmental compartments (e.g., atmosphere, surface water, ocean, soils, etc.), but only the analysis of these chemicals in wildlife can offer a real and precise picture of the amount of chemicals that is really absorbed by them, which therefore exerts adverse effects on health. Biological monitoring, i.e., the analysis of chemicals or metabolites in biological samples, is therefore a useful tool to investigate the body burden and potential health effects of mercury in wildlife.

The toxic effects of this pollutant on wildlife have been studied in different ecosystems, which are varied and complex and include physiological, neurological, behavioural, and reproductive harm causing chromosomal alteration, endocrine disruption, or effects on the immune system, among other impacts [4,8,9].

In this context, and although Antarctica can be considered one of the few environments still unspoiled worldwide, specific incidents and the increasing tourism, together with global pollution, are increasing contamination and adding pressure to the equilibrium and conservation of this unique ecosystem [10,11]. Therefore, there is a need for monitoring the extent of contamination in the Antarctica in order to protect its environment and wildlife. In this regard, Antarctic penguins can be considered as a standard biological indicator for monitoring the contamination of Antarctic nearshore ecosystems, since they have permanent ecological niches and represent an important part of the avian biomass in the region [12,13]; they also have a high trophic level for studying bioaccumulation and biomagnification phenomena.

In general, blood is considered the best matrix for biological monitoring of the exposure to the majority of chemicals [14]. However, it presents limitations in terms of sample accessibility, sample amount, sample preservation, etc., making the possibility of using non-invasive matrices such as feathers highly valuable. This biological matrix can be collected with minimum disturbance to birds and can be stored during years without special conditions (no refrigeration required) in small facilities. Feathers can be a useful biological indicator, especially for mercury, which is absorbed, distributed throughout the organism, and incorporated from the blood to the feathers where it is bound to sulphur groups in the amino acids of the keratin protein [15,16]. This process occurs during the entire process of feather growth [17]. Penguins annually moult all their feathers in a short period of time in which they remain fasting ashore [18], which minimises the potential within-individual variation in mercury determination from feathers [19]. Since mercury present in feathers is up to 80–100% in the form of methylmercury [20], the analysis of total mercury in feathers can be used as a surrogate for methylmercury analysis, significantly simplifying the assessment of mercury exposure.

This study investigated the exposure to mercury in three species of penguins, gentoo (*Pygoscelis papua*), chinstrap (*P. antarcticus*), and Adélie (*P. adeliae*), from seven different locations in a NE–SW gradient from high to low human activity throughout the Antarctic Peninsula area. Potential higher levels of mercury in the northern locations, which have the higher human impact [21], were evaluated. Moreover, considering the expected biomagnification of mercury in the marine food webs [22] and differences in the trophic position of the penguin species studied [23,24] we also evaluated potential differences in mercury levels among the different species.

## 2. Material and Methods

During the 2005/2006 and 2006/2007 austral summer seasons, feather samples (5–10 feathers per individual) were collected from the flanks of adult penguins in seven different locations throughout the Antarctic Peninsula area and individually stored in polyethylene bags. Penguins (*n* = 108) were captured during moulting using a long-handled net, sampled, and immediately released. The sample size for each location and species is shown in Table 1. The populations sampled were located in the southern distribution range of chinstrap and gentoo penguins and in the northern distribution range of the Adélie penguin.

Samples were analysed using the analytical method described by Jerez et al. [13] with minor modifications. In brief, feathers were carefully rinsed up to three times with double-distilled and deionized water to eliminate adsorbed external contamination and dried at 75–80 °C until a constant weight (mean water content of penguin feathers: 11%). Around 0.40 g of the dried and chopped material, according to availability, was subjected to microwave digestion with HNO_3_ (65%), H_2_O_2_ (30%), and H_2_O in the proportion of 5:2:3. Hg concentrations were measured by inductively coupled plasma mass spectrometry (ICP-MS Agilent 7500 ce). The system included a CETAC ASX-510 autosampler, a Peltier-cooled Scott-type nebulizer chamber, a MicroMist concentric nebulizer, nickel cones, a 27.12 MHz radio frequency generator, a 1600 W Fassel-type quartz torch, an argon mass flow control in plasma, an auxiliary line, an adjustment line and carrier gas, a hyperbolic quadrupole mass filter (3 MHz and 2–260 amu), a simultaneous digital/analog detector with 9 orders of magnitude of linear dynamic range, and a collision/reaction cell.

In order to avoid contamination, all the materials used were carefully washed in HNO_3_ (10%) for 24 h, including a last rinsing step with double-distilled and deionized water, final ultrasonic cleaning, and subsequent oven-drying (70 °C). All reagents used were Suprapur (Merck), and the water was double-distilled and deionized (Milli-Q system, Millipore, USA). Quality control of the procedure (reproducibility and reliability of the results) was carried out by introducing the samples in duplicate at random, with blanks at the beginning of each series of analyses and at every 5 samples. The calibration standards were analyzed initially and periodically. Certified reference materials were used (DORM-2 and DOLT-2, Table 2). The seven standards used were analyzed from commercial stock solutions (Merck^®^), which were stabilized with 20 μL of HNO_3_. The concentrations of the standards were 1, 5, 10, 25, 50, 100, and 200 ng/L. In cases where the upper limit of the calibration line was insufficient to detect levels higher than this in the test samples, standards of a higher concentration or dilution of the sample in question were prepared. The correlation index (R2) of the calibration lines was equal to or greater than 0.999.

The detection limit was calculated by using the formula DL = 3 sB/a (DL: detection limit; sB: standard deviation of the number of counts corresponding to zero on the calibration line; a: the constant of the calibration line). According to Smith et al. [25], values below the detection limit (LD) were predicted from expected normal scores when more than 50% of all samples showed detectable levels within each data set. If that was not the case, detection limit values were assigned.

Statistical analyses were conducted using Statistica package version 7. Variables were normally distributed (Kolmogorov–Smirnov test > 0.20). Inter-specific and geographical differences were tested by using one-way ANOVAs (with Bonferroni post-hoc tests) and Student′s *t*-tests. All statistical analyses were two-tailed with a significance level of 0.05. Inter-specific differences were tested in those locations where two or three species shared the same area, whereas geographical differences were tested for each species separately.

## 3. Results and Discussion

Concentrations of Hg in the feathers of gentoo (*P. papua*), chinstrap (*P. antarcticus*), and Adélie penguins (*P. adeliae*) and the study of geographical differences are shown in Figure 1, Figure 2 and Figure 3, respectively. Detectable Hg levels were found in all feather samples of *P. papua* and *P. adeliae* and in the 82% of the samples of *P. antarcticus*. As mentioned before, mercury present in feathers was mainly in the form of methyl-Hg (percentages higher than 80% in aquatic bird feathers) [20], so the analysis of total mercury can be considered a good proxy of the methyl-Hg exposure.

Geographical differences in Hg concentrations were not statistically significant for gentoo penguins (Figure 1) and chinstrap penguins (Figure 2), which does not support for these species the hypothesis of a higher presence of Hg in northern populations. This could be a consequence of the limited sample size of this study. However, chinstrap penguin samples from King George Island showed the highest mercury levels found for this species in all the studied locations, coinciding with what is described by Brasso et al. [26]. Indeed, the highest mercury concentration of all studied penguins was observed in one chinstrap penguin from King George Island with 12,529.8 ng g^−1^. The greater proximity of King George Island to South America, and therefore to anthropogenic sources of mercury, as well as a higher human activity intensity in this location could explain this result; although, 70% of samples in this location showed mercury levels below the detection limit (LD). On the contrary, the results for Adélie penguins seem to support our prediction as the highest levels were found in King George Island; although, differences with the other sampling sites were only marginally significant (Figure 3).

This weak relationship between the level of mercury concentration and a latitudinal gradient for the three species could indicate a homogeneous presence of mercury along the Antarctic Peninsula, but it should be confirmed with further investigations since geographical differences cannot be totally ruled out. These results coincide with Brasso et al. [26], who analysed eggshells of the same three species of penguins, finding that mercury concentrations were fairly homogeneous throughout the Antarctic Peninsula, which suggests little spatial variation in the risk of exposure to dietary mercury in this food web.

Regarding interspecific differences, we found that chinstrap penguin samples contained the highest Hg levels in two locations (Livingston I. and Ronge I., Figure 4), while we did not find interspecific differences in King George I. These results do not support our prediction that a higher trophic niche could be related to higher mercury level due to bioaccumulation and biomagnification processes, as the gentoo penguin usually has a diet including more fishes and cephalopods, therefore occupying a higher trophic niche, than the chinstrap penguin, which shows a diet with a higher percentage of krill [23,24,27]. This coincides with what was described by Brasso et al. [26], which affirms that inter- and intra-specific differences in eggshell membrane mercury concentrations of the same three species of penguins were not related to eggshell δ15N or δ13C values.

Differences in habitat use pelagic/offshore in the chinstrap and benthic/inshore in the gentoo and the higher krill consumption of chinstrap [27] could explain the differences found. Another explanation could involve physiological differences among species such as different capacities for detoxification and elimination of Hg or different absorption-elimination rates.

Hg concentrations in penguin feathers in this study were four to six orders of magnitude higher than those found in inorganic samples such as snow from different Antarctic areas (0.04–0.43 ng g^−1^ in Brooks et al. [28]; 0.0004–0.00096 ng g^−1^ in Sheppard et al. [29]). This is in agreement with the higher mercury concentration found in feathers of Antarctic seabirds in comparison with sediment samples in the same locations [30]. Besides, Hg levels were one to three orders of magnitude higher than those found in Antarctic krill or fish (<0.1–34.6 ng g^−1^ d.w.; [31,32]). These results suggest bioconcentration and biomagnification of Hg in Antarctic top predators. Despite that, the levels detected in this study were lower than those considered toxic for seabirds (around 15,000 ng g^−1^) [33].

Comparing our results with those described in the feathers of other seabirds from different regions, we found that some chinstrap penguins from King George Island showed high Hg levels, similar to those found in Arctic seabirds (average levels between 891.00 and 4855.00 ng g^−1^ d.w. [20,33]). The feathers of chinstrap penguins even showed extremely high individual levels (maximum level of 12,529.8 ng g^−1^ d.w.), although large inter-individual differences were observed, which could perhaps be influenced by the size or age of the individuals [34]. The other two species of penguins studied, gentoo and Adélie, also showed high individual levels and were similar to those found in other seabirds from the Atlantic Ocean, the Indian Ocean, or the Mediterranean Sea (DL-1911 ng g^−1^ d.w. [35,36,37]). One limitation of this study is that, due to sampling conditions, it was not possible to obtain data of the age or gender of all specimens, which could have been very valuable for the interpretation of the results [34].

High Hg levels found in penguin feathers confirm the idea that, during spring and summer, the coastal Antarctic ecosystems can be affected by similar or higher Hg deposition rates than many other regions in the world. Although this phenomenon has been more widely studied in the Arctic, some studies have also confirmed the existence of Hg depletion events in Antarctica [38,39] as well as methylmercury bioaccumulation and magnification in the Antarctic food webs [10,22]. Concentrations of reactive gaseous mercury as high as those found in some industrial environments have been detected in Antarctica after polar sunrise [31]. However, re-emission is probably lower during the dark and cold Antarctic winter since it mainly depends on temperature and microbial activity. It has been estimated that only 10–20% of the deposited Hg is re-emitted [31]. For this reason, Antarctic organisms are especially susceptible to Hg accumulation.

Today these processes achieve special relevance in Antarctica because the use of Hg, among other pollutants, is increasing in the Southern Hemisphere and Asia, contrary to what is happening in North America or Europe. Mercury in the environment is a global problem that requires global solutions such as the Minamata Convention (http://www.mercuryconvention.org/ (accessed on 17 July 2021)). However, despite the current efforts to reduce the use of mercury and protect the environment as well as human populations, the re-emission and the long-range atmospheric transport of this element make it that mercury levels in the environment will not decrease on a short time scale. According to Bargagli [31], this increase could be affecting the Antarctic region, where atmospheric Hg levels have been growing during the last two decades. In addition, the percentage of bioavailable Hg in Antarctica increases in spring when organisms, including penguins, resume their metabolic and reproductive activity. This situation could be even worse as a consequence of global climate change. Some studies have predicted an increase in atmospheric mercury concentrations and in mercury methylation related to rising temperatures [40,41,42].

## 4. Conclusions

Most of the analyzed feather samples showed detectable mercury levels, clearly supporting previous findings of the relevant presence of this environmental pollutant in the Antarctic ecosystem. Even though these results can be related to natural phenomena (e.g., local volcanism), they can also be related to anthropogenic pollution in the region. The latter would explain the high Hg levels detected in the feathers of chinstrap and Adélie penguins collected in King George Island, one of the Antarctic locations with greater human activity and the nearest to South America. This fact could be especially relevant in top predators such as penguins, since the results support the existence of mercury bioconcentration and biomagnification in the Antarctic trophic chain.

Although the levels detected in this study are lower than those considered toxic for seabirds, they are similar or even higher than those found in seabirds from other regions of the world, supporting the hypothesis of mercury depletion, bioaccumulation, and biomagnification in Polar Regions.

Since climate change and rising temperatures can exacerbate the situation by increasing mercury atmospheric concentrations and methylation, this could increase the risk for wildlife in this unique and mercury-sensitive environment. Thus, the control of these pollutant levels present in the Antarctic fauna, and especially in top predators, is crucial for monitoring potential adverse effects.

Penguin feathers are a biological matrix that is particularly useful for monitoring Hg levels and investigating how diet and other factors are related to the levels found; they also support a better understanding of spatial and temporal trends.

## Figures and Tables

**Figure 1 ijerph-18-09918-f001:**
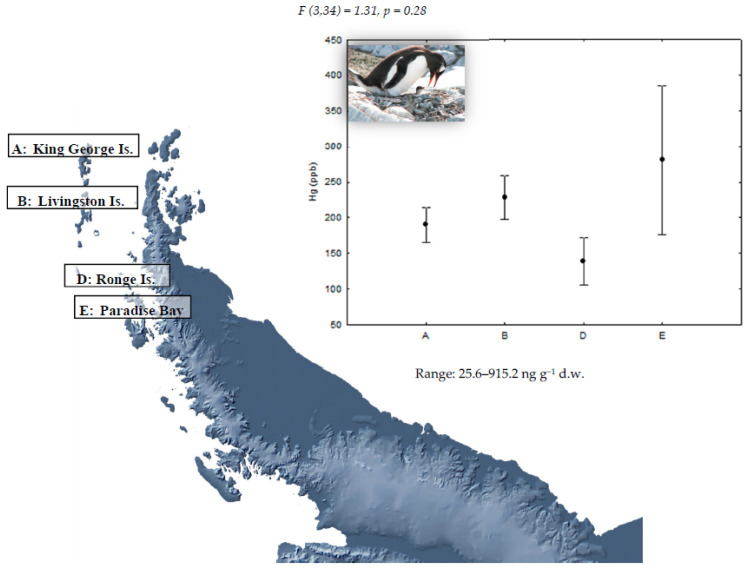
Mean (±SE) Hg concentration in feathers of gentoo penguins (*P. papua*) in four locations of the Antarctic Peninsula area. For sample sizes, see Table 1.

**Figure 2 ijerph-18-09918-f002:**
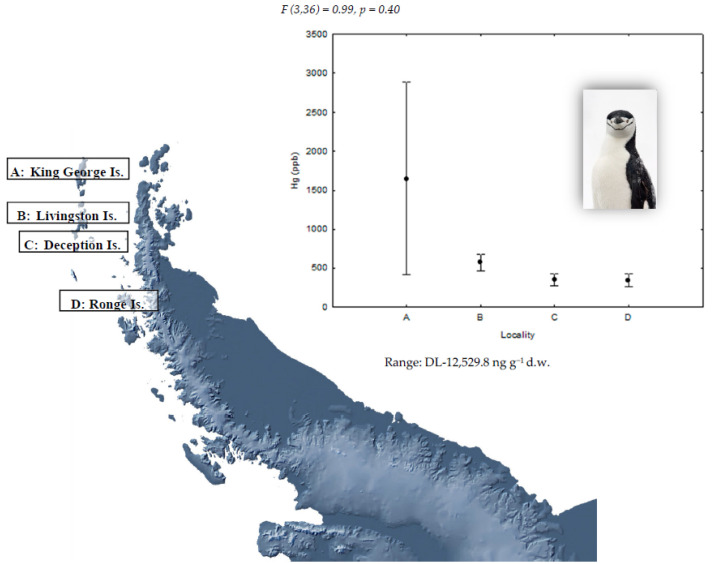
Mean (±SE) Hg concentration in feathers of chinstrap penguins (*P. antarcticus*) in four locations of the Antarctic Peninsula area. For sample sizes, see Table 1. DL: detection limit.

**Figure 3 ijerph-18-09918-f003:**
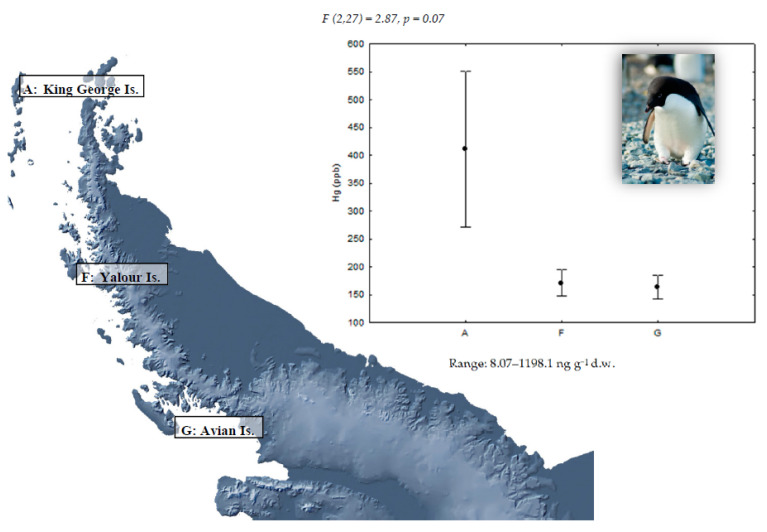
Mean (±SE) Hg concentration in feathers of Adélie penguins (*P. adeliae*) in three locations of the Antarctic Peninsula area. For sample sizes, see Table 1.

**Figure 4 ijerph-18-09918-f004:**
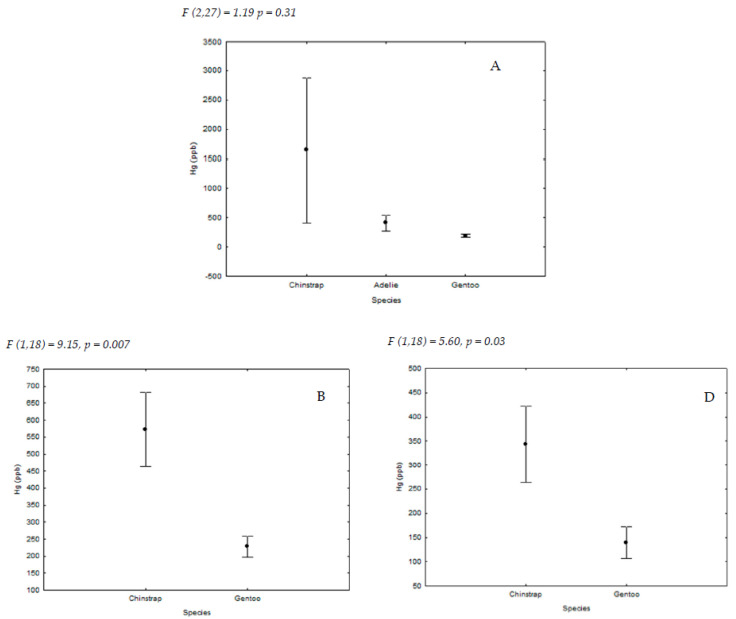
Mean (±SE) Hg concentration in feathers of the three studied species from the islands of (A) King George, (B) Livingston, and (D) Ronge. For sample sizes, see Table 1.

**Table 1 ijerph-18-09918-t001:** Number of individuals sampled in every location for the three penguin species studied.

Locations	Species		
	*P. papua*	*P. antarctica*	*P. adeliae*
King George Island (62°15′ S, 58°37′ W) A	10	10	10
Livingston Island (62°39′ S, 60°36′ W) B	10	10	-
Deception Island (63°00′ S, 60°40′ W) C	-	10	-
Ronge Island (64°40′ S, 62°40′ W) D	10	10	-
Paradise Bay (64°53′ S, 62°53′ W) E	8	-	-
Yalour Island (65°15′ S, 64°11′ W) F	-	-	10
Avian Island (67°46′ S, 68°64′ W) G	-	-	10

**Table 2 ijerph-18-09918-t002:** Detection limit value (ng g^−1^), reference materials values (µg g^−1^), and percentages of recovery obtained.

Element	Detection Limit	DORM-2	% Recovery	DOLT-2	% Recovery
Hg^202^	0.18	4.64 ± 0.26	89.2	2.14 ± 0.28	91.6

## Data Availability

Data available on request due to restrictions privacy. The data presented in this study are available on request from the corresponding author. The data are not publicly available due to restrictions privacy.

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
