# Peer review of "Mercury Levels in Feathers of Penguins from the Antarctic Peninsula Area: Geographical and Inter-Specific Differences"

_ijerph, 2021, doi:10.3390/ijerph18189918_

Round 1

Reviewer 1 Report

The paper “Mercury levels in feathers of penguins from the Antarctic Peninsula area: geographical and inter-specific differences” (Manuscript ID ijerph-1344378) is original and interesting, relevant to the journal thematic, presenting the results obtained by the research team during two periods of mercury monitoring using as biological indicators the feathers of Antarctic penguins, which represent an important part of the avian biomass in the target area. 

The conclusions chapter should be completed with a synthesis of these original results and their interpretation. 

Some references should be completed with DOI and the text in the manuscript should be corrected in several lines before acceptance, such as:

1). Line 205: to replace “inter-specific” with “interspecific”;

2) line 278- please rephrase: …..seems to affect to Antarctica…. 

Thus, I recommend its publication in the journal “International Journal of Environmental Research and Public Health” after minor revision.

Reviewer 2 Report

Congratulations for the paper. Comments and suggestions in the pdf file.

Reviewer 3 Report

This manuscript “ Mercury levels in feathers of penguins from the Antarctic Pen-2 insula area: geographical and inter-specific differences” is based on very few ground-based measurements of Mercury with a very simple analysis. Data and methods and results & discussion sections seem to be very poor. The author should address the following comments:

Line 22: Remove dot (.) after 93%

Line 38: Check superscript for 106-107

Lines 81-87: The author should explain why this study is important.

Please check Table 1, for example, degree, a hyphen, and space

Line 98: Check 75-80ºC. Do it for the entire manuscript.  

What do you mean by Hg202?

Line 123: How have you calculated significance based on the Geographical differences in Hg Concentrations? In addition, your data are very few.

Data and methods: Explain details about some statistical analysis, for example, ANOVA test (regression methods or), how calculated significance?

Results & discussion: This part is not written well, recommending improving this section. Do more analysis.  Actually, this study is very simple.

Round 2

Reviewer 3 Report

The authors have significantly improved the manuscript, now it can be considered for publishing.